# Computational insights into flavonoids inhibition of dengue virus envelope protein: ADMET profiling, molecular docking, dynamics, PCA, and end-state free energy calculations

Amar Waiba[1], Anuraj Phunyal[1], Tika Ram Lamichhane[2], Madhav Prasad Ghimire[2], Hari Nyaupane[1], Ashish Phuyal[1], Achyut Adhikari[1]*

1 Central Department of Chemistry, Tribhuvan University, Kirtipur, Kathmandu, Nepal, 2 Central Department of Physics, Tribhuvan University, Kirtipur, Kathmandu, Nepal

☯ These authors contributed equally to the work.
* achyutraj05@gmail.com

## Abstract

Dengue virus is a critical worldwide health concern, and efforts to identify useful antiviral drugs remain imperative. This study utilized computational techniques to investigate the flavonoids as a potential inhibitor of the dengue virus envelope protein (PDB ID: 1OKE). 33 flavonoids were docked among them, 5-hydroxy-3-(4-hydroxyphenyl)-7-[(2S, 3R, 4S, 5S, 6R)-3,4,5-trihydroxy-6-(hydroxymethyl)oxan-2-yl]oxychromen-4-one (FLA1) showed the best binding affinity of -9.1 kcal/mol towards the E protein. Molecular dynamics simulations (100 ns) were carried out to analyze the stability and interaction of protein-ligand complexes, including parameters such as RMSD (FLA1 of $2.36 \pm 0.43$ Å), RMSF, Rg, SASA, hydrogen bonding, and RDF. In addition, PCA and DCCM analysis exposed considerable conformational differences and residue correlations favoring FLA1 stability. The binding free energy calculations using the MM/PBSA methodology confirmed the strong binding ($-29.1 \pm 5.83$ kcal/mol) of FLA1 to the target protein. ADMET profiling also revealed good pharmacokinetic properties. These findings suggest FLA1 is a possible inhibitor of the dengue virus and a promising drug candidate for the development of antiviral drugs in the future.

## 1. Introduction

An infected mosquito transmits Dengue virus (DENV) (pathogen of the *Flaviviridae* family) and is responsible for widespread infection globally each year, particularly in tropical and subtropical areas [1–3]. The estimated global economic burden of dengue is around USD 8.9 billion annually [4]. Thus, it is crucial to prevent and treat these arboviral infectious diseases [5]. The range of health effects of DENV infection includes dengue fever (DF), which progresses into dengue hemorrhagic fever (DHF) in some patients, and in severe cases, dengue shock syndrome (DSS) [6].

**Data availability statement:** All relevant data underlying the finding of this study are now publicly available via Figshare repository https://doi.org/10.6084/m9.figshare.29205689.v1; https://doi.org/10.6084/m9.figshare.29206427.v1 for the replicate the results reported in this article.

**Funding:** The author(s) received no specific funding for this work.

**Competing interests:** No authors have competing interests.

At present, no specific antiviral treatment is available for dengue. Under medical supervision, people with dengue fever should get enough sleep, drink lots of fluids to stay hydrated, and take paracetamol to treat their illness. Though there are abundant possible antiviral candidates, only a few, such as celgosivir, lovastatin, prednisolone, balapiravir, and chloroquine, have undergone clinical trials [7,8]. There is currently only one approved dengue vaccine (CYD-TDV, Dengvaxia), developed by Sanofi Pasteur and comprising four shorts in total (CYD-1–4), which is used in some countries [9,10]. However, concerns about its general effectiveness among people [11]. Therefore, a pressing need remains to explore new therapeutic candidates that target the dengue virus.

The genome of DENV comprises a single positive-sense strand of whole Ribonucleic Acid (RNA), measuring 11 kb in size. The RNA genome consists of an individual open reading frame (ORF) bordered by untranslated places (UTRs) [12]. This ORF encodes a polyprotein, which is later cleaved into proteins that are structural and nonstructural; the structural proteins include pre-membrane (prM), envelope (E), and Core or capsid (C), while the nonstructural proteins consist of NS1, NS2A, NS2B, NS3, NS4A, NS4B, and NS5. Nonstructural proteins are crucial for processes such as viral RNA replication and evading the host immune system. In contrast, structural proteins are responsible for functions such as receptor binding, fusion, maturation, and assembly of DENV [13,14]. E protein consists of 394 amino acid residues made from the three structural parts, Domain I, II, and III [15]. Domain II (DII) contains the fusion peptide loop, which facilitates E to insertion into the host cell membrane, but when pH alters, its structural integrity where Domain III (DIII) plays a role in receptor binding (Fig 1) [16].

The dengue virus E protein's complicating structural changes stimulate the joining of the viral envelope and cell membranes. Initially, the acidic environment of the endosome and a specific histidine residue in E undergo protonation, prompting the reversible separation of protein dimers of the virion surface [17,18]. This procedure, made possible by DII's flexible movement around a hing domain at Domains I and II intersections, exposes the fusion peptide loop at the end of Domain II (DII) and permits entry into the host cell membrane [19–21]. During this transformation, the E protein acts as a bridge that connects the viral envelope with the cellular membrane, and the homotrimers of E proteins are formed permanently [22,23]. Thus, the E protein in the design of DENV entry inhibitors is one of the more promising methods for

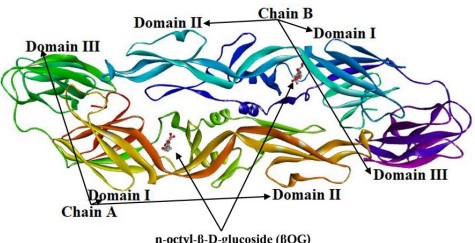

**Fig 1. Cartoon representation of envelope protein showing chain A, chain B, Domains, and βOG.**

inhibiting the virus. In recent *in silico* screenings, most inhibitors target the n-octyl-β-glucoside (βOG; native ligand) pocket as a binding site [20].

Flavonoids are polyphenolic compounds found in several plant species with various properties, such as antiviral, antioxidant, antibacterial, and antimutagenic [24–27]. Previous studies have reported several flavonoids as inhibitors of dengue through various mechanisms. However, a systematic evaluation of flavonoids targeting the DENV E protein remains limited.

In this study, we made a database of flavonoids (*ca.* 3000 compounds) based on literature reports on their antiviral activity, structural diversity, and drug-likeness profiles [28–30]. The main goal was to explore the inhibitory effects of flavonoids against an envelope inhibitor among the tested molecules. By utilizing molecular docking, molecular dynamics simulations, and free energy calculations, we predict where the flavonoids bind to the E protein structure, assess their interaction and stability in the orthosteric site, and determine the drug-likeness properties of flavonoids. The compound FLA1 emerged as a promising hit for further investigation. This work provides preliminary computational insights into flavonoid-based E protein inhibition and lays the groundwork for future *in vitro* and *in vivo* validation.

## 2. Material and methods

### 2.1. Selection and preparation of ligand database

A curated database of compounds (*ca.* 3000 flavonoids) was obtained from different literature sources. The 3D structures of these compounds retrieved from the PubChem website (https://pubchem.ncbi.nlm.nih.gov/) in SDF format. The ligands preparation involved adding polar hydrogens, energy minimization, and verification of molecular formulas using the Avogadro software v1.1 [31]. The resulting structures were then converted from SDF to PDB fotmat using PyMOL v2.5.4 for molecular docking studies [32].

### 2.2. Protein preparation and validation

The DENV E protein was selected as a target, corresponding to PDB ID 1OKE (Expression organism: *Drosophila melanogaster*) (https://doi.org/10.2210/pdb1OKE/pdb) [20]. Its crystal structure, which has 394 amino acid residues and a resolution of 2.40 Å, was obtained from the Protein Data Bank web server (https://www.rcsb.org/). The structure includes two chains (A and B), with chain A selected for further studies. PyMOL v2.5.4 was employed to remove chain B, water molecules, and co-crystallized ligands, and adding hydrogen atoms to prepare the protein structure for analysis. Subsequently, AutoDockTools v1.5.7 introduces a Kollman charge of -84.799, ensuring system neutrality and converting the PDB to PDBQT format [33]. The structure analysis and validation of the protein using the SAVES v6.0 web server (http://saves.mbi.ucla.edu/) provides three different applications: ERRAT, VERIFY 3D, and PROCHECK [34].

### 2.3. ADMET prediction

The multiple web servers (ADMETlab2.0, ProTox-II, pkCSM, and swissADME) were used to evaluate the pharmacokinetic and pharmacodynamic properties of compounds [35–38].

### 2.4. Molecular docking

The Autodock Vina software v1.2 was used for molecular docking, which is based on the Lamarckian Genetic Algorithm (LGA) and used to generate AutoGrid and grid maps with X = 40 Å, Y = 40 Å, and Z = 40 Å as typical grid box sizes, the energy range of 3, spacing of 0.375, and 20 modes with exhaustiveness of 32, the grid center dimensions set at X = -12.964, Y = 80.186, and Z = 45.884, after that, perform docking on the βOG pocket (orthosteric site) and confirmed active sites of proteins using CASTp server [39,40]. The top three protein-ligand complexes (according to their binding affinity) were saved in PDB format and then converted into PDBQT for molecular dynamics simulations. Visualizing the binding interaction between protein and ligand using the Discovery Studio 2021 program [41].

## 2.5. Molecular dynamics simulation

The GROMACS program simulated the best binding affinity complexes obtained after molecular docking [42]. The Charmm27 force field was used for receptor topology due to its compatibility with ligand topology files [43]. The SwissParam web server (http://swissparam.ch/) was used for ligand force field in the.zip format [44]. For solvation of the system, the TIP3P water model within a triclinic box (a = 5.86 nm, b = 7.34 nm, c = 14.39 nm) with a spacing of 10 Å at the sides was chosen to prevent interactions between the periodic images. The system was neutralized by adding a 0.15 M NaCl solution. Equilibration was executed in four phases, each lasting 200 ps, which were used in the equilibration procedure, which was carried out at 310 K, the physiological temperature. In the first two stages, the equilibrium was reached at constant temperature and volume (NVT), while the subsequent two steps attained equilibrium under constant pressure and temperature (NPT). During equilibration, temperature coupling was utilized for the modified Berendsen thermostat, and pressure coupling was performed using the Berendsen method. Particle Mesh Ewald (PME) was utilized to manage distant coulomb interactions [45]. After equilibration, a 100 ns production run was carried out without restraints with a step size of 2 fs. Using inbuilt modules, several geometrical parameters were tracked during these simulations, including Root Mean Square Deviation (RMSD), Root Mean Square Fluctuation (RMSF), Radius of Gyration (Rg), Solvent Accessible Surface Area (SASA), Hydrogen bond count, Radial Distribution Function (RDF), Principle Component Analysis (PCA), and Dynamic Cross-Correlation Matrix (DCCM).

## 2.6. Binding free energy calculation

The gmx_MMPBSA module of GROMACS computes the binding free energy of the complexes over the final 10 ns equilibrated trajectories and was thoroughly described in Phunyal et al. [46].

## 2.7. DCCM and PCA calculations

The R programming package Bio3D was used to perform DCCM and PCA analysis from MD trajectories. A detailed protocol was outlined at Gyawali et al. [47].

## 3. Results

### 3.1 Validation of protein structure and analysis

Before conducting molecular docking, the E protein underwent preparation and validation through server analysis. The protein's ERRAT score was determined to be 84.6154, showing favorable quality, while from the VERIFY module, its structural integrity was 82.23%. Based on a complex deposited in a protein data bank, analysis of the active site identified **GLN271, GLU49, THR48, ALA50, LEU198, ILE270, PHE193, LEU207, PHE297, THR280, VAL130, LEU135, GLN200, SER274, ALA205, and LEU277** amino acid residues interacting via hydrogen bonding, Van der Waals', Carbon hydrogen bond, Alkyl, pi-Alkyl. The Ramachandran plot from the PROCHECK module of the SAVES server illustrates the configurations of phi (φ) and psi (ψ) angles within a polypeptide. Approximately 81.8% of residues occupy the most favored regions on this plot (S1 Fig). Another 17.6% fall within the additional allowed region, leaving only 0.6% in the disallowed regions. This distribution suggests a well-constructed protein model with high-performance expectations.

### 3.2 Validation of molecular docking

The native ligand was re-docked to the protein, and less than 1.912 Å of RMSD relative to the co-crystal pose in the crystal structure showed molecular docking was valid [48]. The parameters were chosen to validate docking for other ligands (Fig 2).

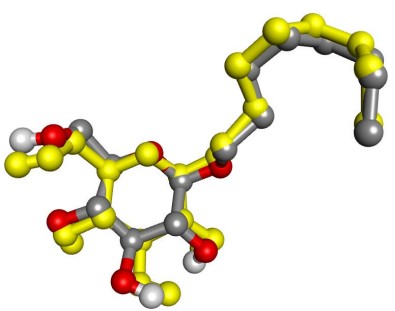

**Fig 2. Docking validation.** Binding interactions between the first docked βOG (grey), sourced from PDB, and the re-docked βOG (yellow) ensure an RMSD of less than 2 Å.

### 3.3 Drug-likeness and ADMET profiling

Out of *ca.* 3000 flavonoids screened, only 33 passed toxicity criteria (hepatotoxicity, carcinogenicity, mutagenicity, immunotoxicity, and cytotoxicity). The hit candidates are classified as class 4–6 and have $LD_{50}$ values over 2000 mg/kg, indicating they have low acute toxicity (S1 Table). Regarding drug-likeness, the physicochemical properties of hit candidates are in an acceptable range and adhere to Lipinski's rule of five (1 violation; FLA1) and Veerber's criteria (S2 Table). Pharmacokinetically, the compounds had a good plasma protein binding (PPB) range between 71% and 98%, suggesting a favorable therapeutic index and prolonged systemic circulation (S3 Table).

Regarding safety, the hit candidates demonstrated acceptable results for skin sensitization (0 to 0.8) and low eye corrosion or irritation, which indicated a low risk of dermal and ocular toxicity (S4 Table). Similarly, blood bank barrier (BBB) permeability value of <-3 logPS < -3 categorizes them as CNS-impermeable. While such CNS non-penetrance could also be a limitation in other applications, if targeting the nervous system is not desired in the treatment of dengue (where CNS effects are undesirable), a limited BBB permeability may be an advantage for reducing off-target neurotoxicity.

Furthermore, the candidates were found to possess a low gastrointestinal (GI) toxicity and were unable to inhibit major cytochrome P450 enzymes (CYP2D6, CYP1A2, CYP1A2, CYP2C19, CYP2C9, and CYP3A4), which suggests low metabolic interaction risk [49]. The flavonoids were also not identified as substrates for renal Organic Cation Transporter 2 (OCT2), suggesting minimal interference in renal clearance mechanisms (S5 Table). Compared to a reference ligand (NITD448) (class 5, $LD_{50}$ = 2150 mg/kg), it showed immunotoxicity, did not follow Lipinski's rule of five, and did not follow Verber's rule. Similarly, it showed greater PPB (99.18%). Therefore, the chosen hit candidates possess an acceptable ADMET profile.

### 3.4 Molecular docking analysis

Molecular docking was used for molecular interactions of different protein-ligand complexes at a molecular level. Using AutoDock Vina, molecular docking was carried out on the βOG pocket of the dengue virus's E protein. This investigation contains 33 flavonoids, native ligands, and NITD448 (reference ligand). We examined these chemical binding affinities with E protein and studied how they interact with hydrophobic areas. When compared to the native ligand (-5.7 kcal/mol) and reference ligand (-8.1 kcal/mol), all 33 flavonoids showed lower binding affinities (ranging from -9.1 to -6.8 kcal/mol) (Table 1).

Among 33 flavonoids, the top three compounds with the highest binding affinities were studied for further calculations (S2 Fig). The molecular details of different flavonoids bonding interactions with key amino acid residues and their distance have been obtained (Table 2). FLA1 exhibits the best binding affinity among the compounds, which have a binding energy of -9.1 kcal/mol. Key amino acid residues including **THR48** (4.21 Å), **ALA50** (3.64 Å), and **THR280** (**3.04** Å) interact with

**Table 1. Ligands with its binding affinity.**

| S.N | Flavonoids (PubChem CID) | Representative name | Binding affinity (Kcal/mol) |
|---|---|---|---|
| 1 | 5281377 | FLA1 | −9.1 |
| 2 | 5281607 | FLA2 | −8.6 |
| 3 | 442520 | FLA3 | −8.6 |
| 4 | 5322064 | FLA4 | −8.6 |
| 5 | 5281616 | FLA5 | −8.5 |
| 6 | 5320315 | FLA6 | −8.4 |
| 7 | 5281611 | FLA7 | −8.4 |
| 8 | 5280443 | FLA8 | −8.3 |
| 9 | 5280863 | FLA9 | −8.3 |
| 10 | 5280666 | FLA10 | −8.2 |
| 11 | 5281612 | FLA11 | −8.2 |
| 12 | 5281628 | FLA12 | −8.1 |
| 13 | 1203 | FLA13 | −8.1 |
| 14 | 12912214 | FLA14 | −8.0 |
| 15 | 5319484 | FLA15 | −8.0 |
| 16 | 5280637 | FLA16 | −7.8 |
| 17 | 5280373 | FLA17 | −7.7 |
| 18 | 11449086 | FLA18 | −7.6 |
| 19 | 5280457 | FLA19 | −7.6 |
| 20 | 5281708 | FLA20 | −7.6 |
| 21 | 182232 | FLA21 | −7.5 |
| 22 | 6453244 | FLA22 | −7.5 |
| 23 | 5280961 | FLA23 | −7.5 |
| 24 | 5320946 | FLA24 | −7.5 |
| 25 | 5281804 | FLA25 | −7.5 |
| 26 | 1226045 | FLA26 | −7.4 |
| 27 | 5282102 | FLA27 | −7.3 |
| 28 | 5281702 | FLA28 | −7.3 |
| 29 | 5272653 | FLA29 | −7.2 |
| 30 | 5322078 | FLA30 | −6.8 |
| 31 | 5468749 | FLA31 | −6.8 |
| 32 | 5281703 | FLA32 | −6.8 |
| 33 | 631095 | FLA33 | −6.8 |
| 34 | 62852 | Native ligand (βOG) | −5.7 |
| 35 | 139031065 | NITD448 (Reference drug) | −8.1 |

FLA1 through hydrogen bonding, while additional interactions like Van der Waals' force of attraction (**ALA205**, **GLN200**, **SER274**, LYS47, **GLU49**, **LEU277**, **PHE193**, LEU191, GLY281, GLY190, **PHE279**, THR268), Pi-Sigma (**LEU 207**) (Fig 3a). FLA2 exhibits a slightly higher binding affinity of −8.6 kcal/mol. Like, FLA2 interacts with hydrogen bonding through **ALA50** (**3.39 Å**), **GLN200** (4.49 Å), and **GLN271** (3.84 Å), amino acid residues. In addition, other interactions such as Pi-Alkyl (**LEU198**, **ILE270**), and Van der Waals' (**PHE193**, **THR280**, **THR48**, **GLU 49**, **LEU277**, ASP203) (Fig 3b). Similarly, FLA3 demonstrates binding affinities similar to FLA2, which is −8.6 kcal/mol. In the case of FLA3, displayed interactions by hydrogen bonding by **ALA50** (**3.34 Å**) amino acid residue. Additional bonds like Pi-Alkyl (**ILE270**, **LEU198**), and Van der Waals' (**PHE193**, **THR280**, **THR48**, **GLU49**, **LEU277**, **GLN271**, **GLN200**) (Fig 3c). Interaction of native ligand

**Table 2. Key amino acids and various interaction types among the top 3 candidates.**

| Compounds PubChem CID (Representative Name) | Interactions | Key amino acid residues with distance (Å) |
|---|---|---|
| | H-bonding | **THR280 (3.04)**, **THR48** (4.21), **ALA50** (3.64) |
| 5281377 (FLA1) | pi-sigma | **LEU207** (4.82) |
| | Van der Waals' | **ALA205, GLN200, SER274,** LYS47, **GLU49, LEU277, PHE193,** LEU191, GLY281, GLY190, **PHE279,** THR268 |
| | H-bonding | **GLN200** (4.49), **GLN271** (3.84),**ALA50** (3.39) |
| 5281607 (FLA2) | pi-alkyl | **ILE270** (4.90), **LEU198** (4.91) |
| | Van der Waals' | **THR48, GLU49, LEU277,** ASP203, **THR280, PHE193** |
| | H-bonding | **ALA50 (3.34)** |
| 442520 (FLA3) | pi-alkyl | **LEU198** (4.95), **ILE270** (4.98) |
| | Van der Waals' | **GLU49, LEU277, GLN271, GLN200, PHE193, THR280, THR48** |
| 62852 (Native ligand) | H-bonding | **ALA50 (3.50)** |
| | pi-alkyl | **LEU207 (4.62), VAL130 (4.92), ILE (4.31)** |
| | Van der Waals' | **ALA205, GLN200, GLU49, LEU277, PHE279,** LEU135, THR280, GLN271 |
| 139031065 (Reference ligand) | H-bonding | THR268 (4.71), GLU269 (3.77) |
| | pi-alkyl, pi-sigma, pi-pi stacked | **LEU207 (4.93), ALA50 (4.44), LEU198 (4.52),** ILE270 (4.34), LEU191 (4.77), PHE279 (4.53) |
| | Van der Waals' | MET196, HIS282, GLY281, **THR48, THR280, PHE193,** GLY190 |

with hydrogen bonding through **ALA50** (**3.50 Å**); meanwhile, extra interactions such as Van der Waals' force of attraction (**ALA205, GLN200, GLU49, LEU277, PHE279**, LEU135, THR280, GLN271) and pi-alkyl (**LEU207, VAL130, ILE270**) (Fig 3d). Likewise, reference ligand exhibits interactions with hydrogen bonding through THR268 (4.71 Å) and GLU269 (3.77 Å) and additional interactions, including Van der Waals' force of attraction (MET196, HIS282, GLY281, **THR48, THR280, PHE193**, GLY190), Pi-alkyl (**LEU207, ALA50, LEU198, ILE270**), Pi-sigma (LEU191) and Pi-Pi stacked (PHE279) (Fig 3e). Previous studies using the same receptor reported similar active site residues, indicating that the calculations were consistent [50,51]. Distinct hydrophobic (non-covalent) and hydrophilic regions were identified for each of the four adducts through 3D interaction analysis (S3 Fig). The different coloured regions signify hydrophilicity (blue, which indicates a higher number of electronegative atoms) and hydrophobicity (brown, which indicates a smaller number of electronegative atoms) parts in the protein structure.

### 3.5 Molecular dynamics simulation analysis

**3.5.1 Root mean square deviation (RMSD).** The protein-ligand complexes' dynamic stability and conformational behaviour were analyzed using the RMSD curve of the ligand and protein backbone (Fig 4a and 4b). Overall, the ligand RMSD values remained below 5 Å (except FLA3), indicating that the ligands bind strongly at the orthosteric site throughout the 100 ns MDS. Notably, FLA1 and native ligand exhibited the lowest average RMSD values of 2.57 ± 0.46 Å and 2.20 ± 0.41 Å, respectively, suggesting a more stable interaction compared to FLA2 of 2.28 ± 0.72 Å, FLA3 of 3.65 ± 0.94 Å, and reference ligand of 3.99 ± 0.76 Å (Fig 4a). The higher fluctuation in FLA3 during the initial 20 ns of MDS, and afterwards, maintains an RMSD of 3.65 ± 0.94 Å.

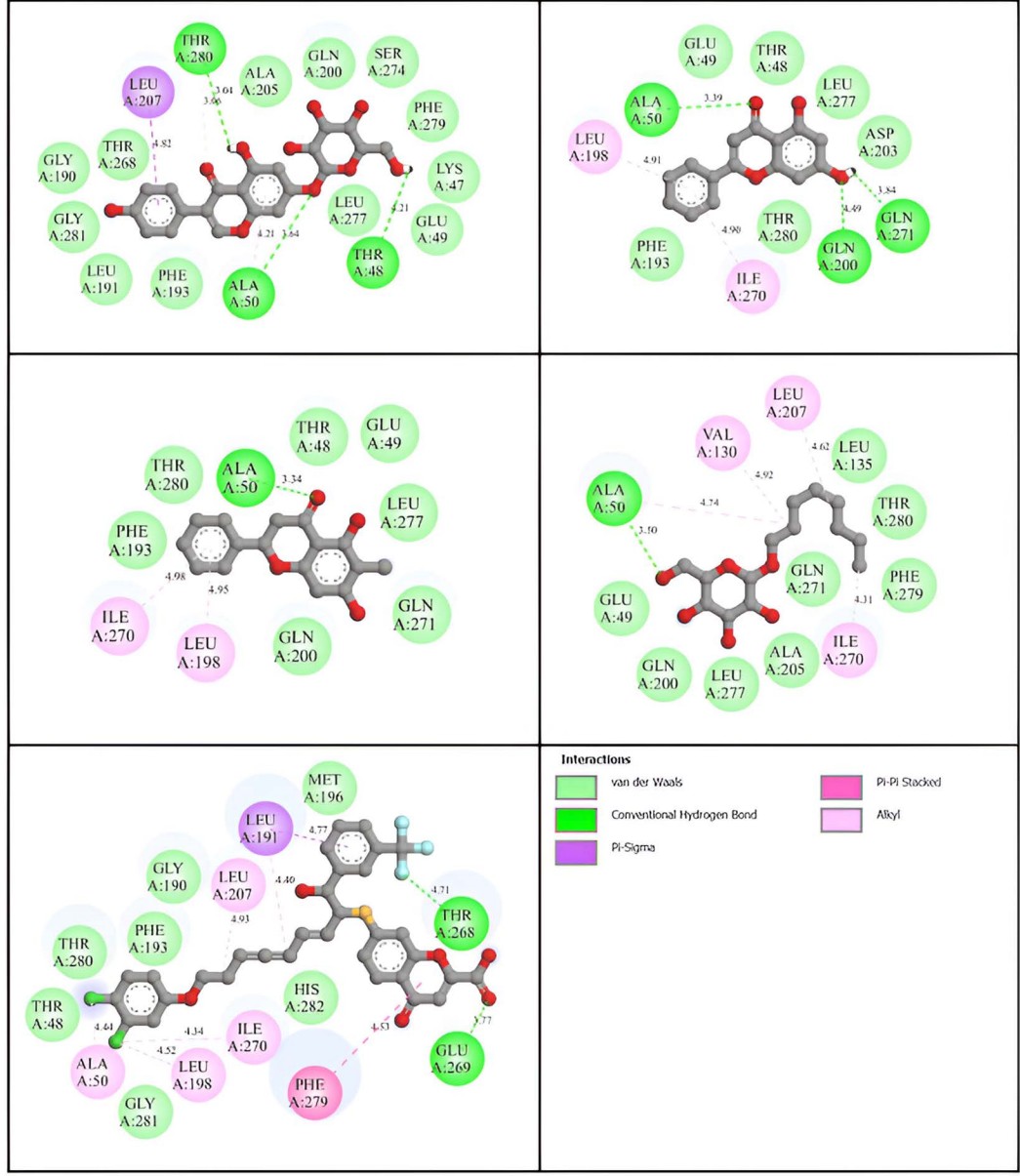

**Fig 3. Two-dimensional projection of the protein-ligand interaction.** (a) FLA1 (b) FLA2 (c) FLA3 (d) Native ligand (e) Reference ligand.

The protein backbone RMSD profiles of protein-ligand complexes showed that protein-FLA1 complexes displayed smooth trajectories with RMSD of 2.36±0.43 Å, Whereas FLA2, FLA3, and native ligand reveal the instability among which FLA3 expresses maximum shifts. Both of them attained stability after 25 ns, with a 3.83±0.96 Å, 5.29±1.74 Å, and 3.36±0.79 Å, respectively. Likewise, the reference ligand shows fluctuation from 0 ns to 50 ns. That was due to the change in conformation of the ligand, and later on, it became smooth with 4.15±0.76 Å (Fig 4b). Similar results were observed during the former work [50]. These differences in RMSD are functionally relevant, as ligand binding into the orthosteric pocket is intended to block the conformational transition of the DENV protein from dimer to trimer. Overall, the lowest RMSD value of ligand FLA1 suggests it can more effectively stabilize the conformation of the E protein.

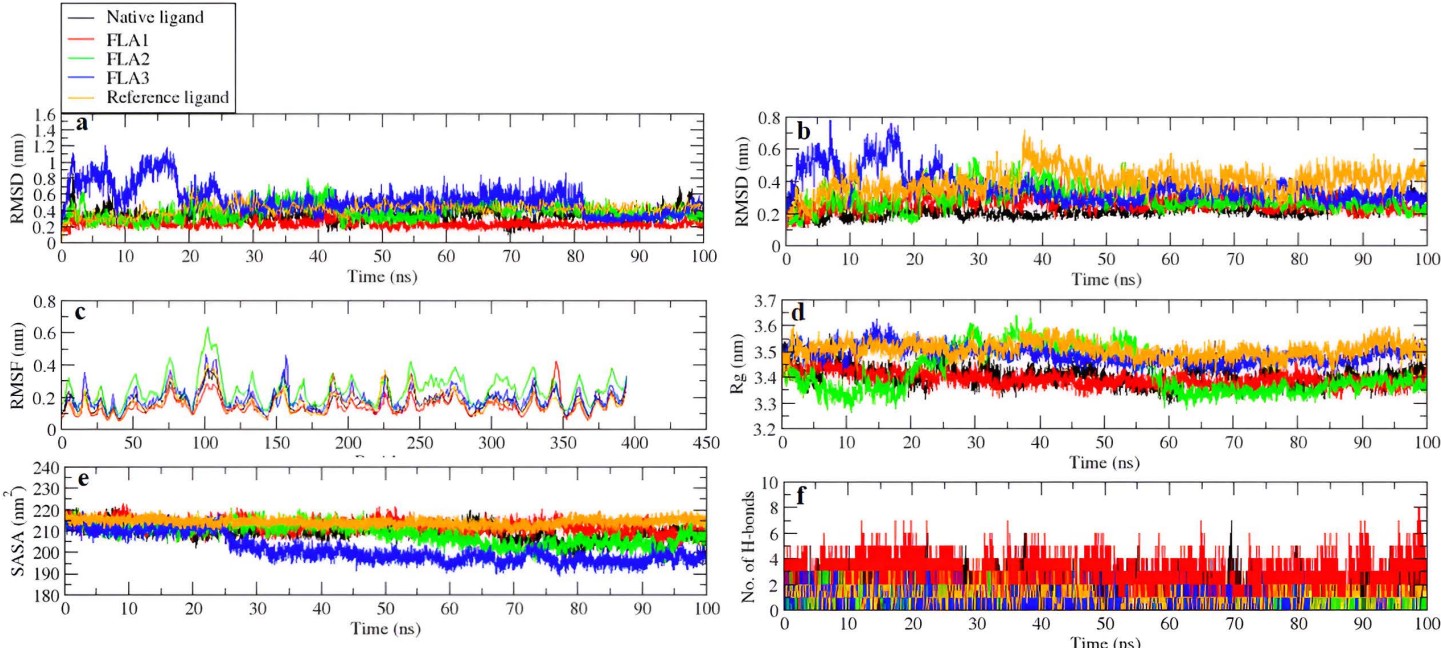

**Fig 4. The RMSD of, (a) the protein backbone relative to the ligand (b) the protein backbone related to the protein backbone, c) The RMSF curves of the protein's α-carbon atoms, (d) The protein's Rg curves, (e) The SASA of the protein, and (f) The count of hydrogen bonds that bind ligands to proteins in various complexes derived from 100 ns MDS trajectories.**

**3.5.2 Root mean square fluctuation (RMSF).** The alpha carbon atoms RMSF of amino acid residues facilitates in evaluating how each amino acid's residues fluctuate when the ligand binds; the more stable the protein geometry, the lower the RMSF value (less than 5 Å) [52]. RMSF plots were generated from MDS for each complex, as shown (Fig 4c). The FLA2 and FLA3 complexes showed a greater fluctuation between 90–115 amino acid residues and a fluctuation of $2.44\pm0.87$ Å and $1.89\pm0.66$ Å, respectively. That is due to the region lacking beta sheet or alpha helix structures and having little to no interaction with the ligand; it seems that these variations have very small to no effect on the instability of the complex. Overall, FLA1, native ligand, and reference ligand show less fluctuation of $1.44\pm0.55$ Å, $1.69\pm0.58$ Å, and $1.63\pm0.62$ Å, respectively.

**3.5.3 Radius of gyration (Rg).** The Rg indicates the compactness of protein-ligand complexes; smaller Rg values indicate a more compact structure [53]. The FLA2 and FLA3 complexes displayed a significant expansion of the protein within an interval of 20 ns to 60 ns afterwards; they both exhibited smooth lines with a value of $34.17\pm0.81$ Å and $34.87\pm0.30$ Å, respectively. In comparison, the native ligand and reference ligand were found to be smooth curves with values of $34.05\pm0.37$ Å and $35\pm0.29$ Å, respectively. The FLA1 ligand complex showed a significant uniform trajectory of Rg of $33.92\pm0.26$ Å, indicating a slight rise of the protein upon ligand binding (Fig 4d).

**3.5.4 Solvent accessible surface area (SASA).** The protein's SASA facilitates identifying the total wettable area for protein throughout MDS. It was found to be $212.17\pm2.7\,nm^2$, $208.52\pm4.63\,nm^2$, $201.49\pm6.45\,nm^2$, $209.87\pm3.64\,nm^2$, and $214.20\pm1.66\,nm^2$ in protein FLA1, FLA2, FLA3, native ligand, and reference ligand complexes, respectively (Fig 4e). The trajectory of the complex formation displayed reduced fluctuation for SASA, suggesting the complex's stability with a constant result in the surface area of the protein interacting with the solvent. The protein does not unfold or fold to indicate structural integrity, and its hydrophobic area is not exposed to the solvent molecules [54].

**3.5.5 Hydrogen bond count.** The total number of conventional hydrogen bonds in protein-ligand complexes affects the stability of the complex (Fig 4f). Stability increased through a rise in hydrogen bonds [55]. FLA1 displayed a maximum number

of eight hydrogen bonds, while the native ligand displays seven. In contrast, FLA2 and FLA3 exhibited three hydrogen bonds, while the reference ligand shows a minimum of two hydrogen bonds. The larger RMSD of FLA2 and FLA3, along with a smaller RMSD of FLA1 and the native ligand, was likely caused by the difference in hydrogen bond formation (Fig 4a).

**3.5.6 Radial Distribution Function (RDF).** Radial distribution functions (RDFs) of hydrogen bonding atom pairs between proteins and ligands observed during molecular dynamics (MD) simulations illustrate both sharp and broad peaks (Fig 5) [56]. Sharp RDF peaks indicated stronger hydrogen-bonding interactions, while broad RDF peaks showed weaker hydrogen bonding [57].

In FLA1, the atom pairs H1(FLA1)-O(THR48) and O2(FLA1)-HN(ALA50) displayed sharp peaks at 2.45±0.56 Å and 2.5±1.45 Å respectively, indicating strong hydrogen bonds. H(FLA1)-OG(THR280) showed a broader and lower peak at 6.85±0.73 Å, suggesting a weaker interaction. For FLA2, the pairs of O1(FLA2)-HN(ALA50) and H1(FLA2)-OE1(GLN271) reveal distinct RDF peaks at 2.5±0.53 Å and 1.95±0.27 Å, respectively, indicating strong hydrogen bonding. Meanwhile,

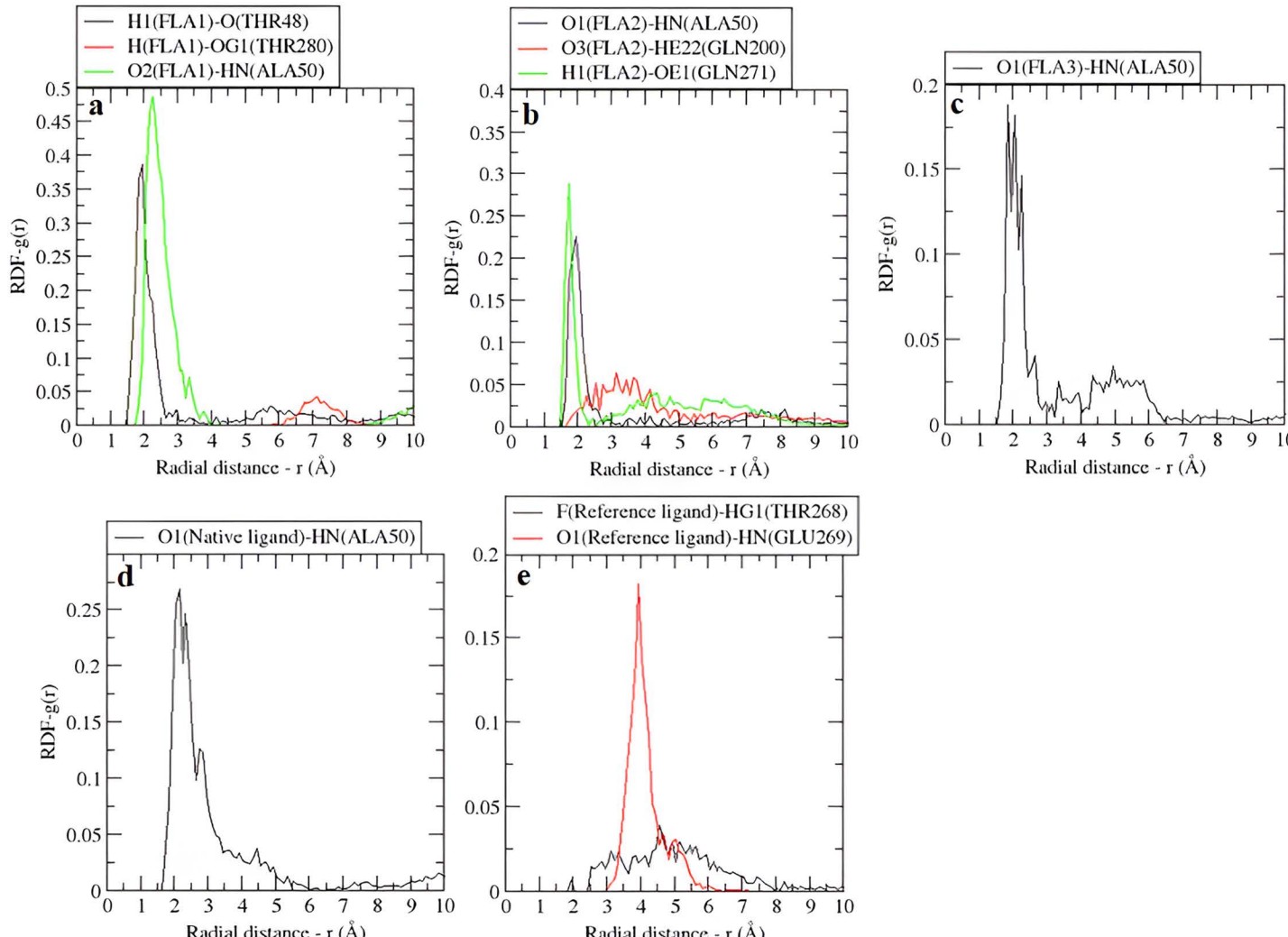

**Fig 5. RDFs of highly interacting atom pairs among the ligand and protein residues that form H-bonds with each other in, (a) FLA1, (b) FLA2, (c) FLA3, (d) Native ligand, and (e) Reference ligand.**

the pair O3(FLA2)-HE22(GLN200) signified a wide and lower peak at 3.9 ± 1.28 Å, indicating weaker interaction. In FLA3, a single interaction between O1(FLA3)-HN(ALA50) produces a sharp yet slightly broad peak at 2.85 ± 0.73 Å, reflecting a moderate hydrogen bond. For the native ligand, the O1(Native ligand)-HN(ALA50) interaction produces a clear peak at 3.65 ± 1.14 Å, consistent with a stable hydrogen bond. Similarly, in the reference ligand, O1(Reference ligand)-HN(GLU269) generates a strong peak at 4.45 ± 0.85 Å, signifying a strong hydrogen bond, while F(Reference ligand)-HG1(THR268) showed a weaker, broader peak at 5.3 ± 1.63 Å, indicating a much weaker interaction. Overall, the FLA1 shows stronger hydrogen bonding with the E protein, to a greater extent, which can be verified from the hydrogen bond count, which has eight hydrogen bonds, which is the maximum among them.

**3.5.7 Ligand snapshots at the protein's active site during simulations.** To investigate the alignment (rotational motion) and location (translational motion) of these docked ligands, snapshots were taken at various instantaneous times of MDS (0, 25, 50, 75, and 100 ns) (Fig 6). With few exceptions, most ligands stayed in the same spot but had different orientations. In FLA1, the ligand exhibits exceptional data in which it remained in the same position as well as with the same orientation in the active site; also, in the case of the native ligand and reference ligand complexes, similar information was obtained but slightly changed in orientation; both of these details valid by smooth backbone protein-ligand RMSD. The ligand in the FLA2-protein complex stays in the same spot but has a different orientation. In the initial phase, the protein backbone in the FLA3-protein complex appears to undergo a substantial conformational shift, as evidenced

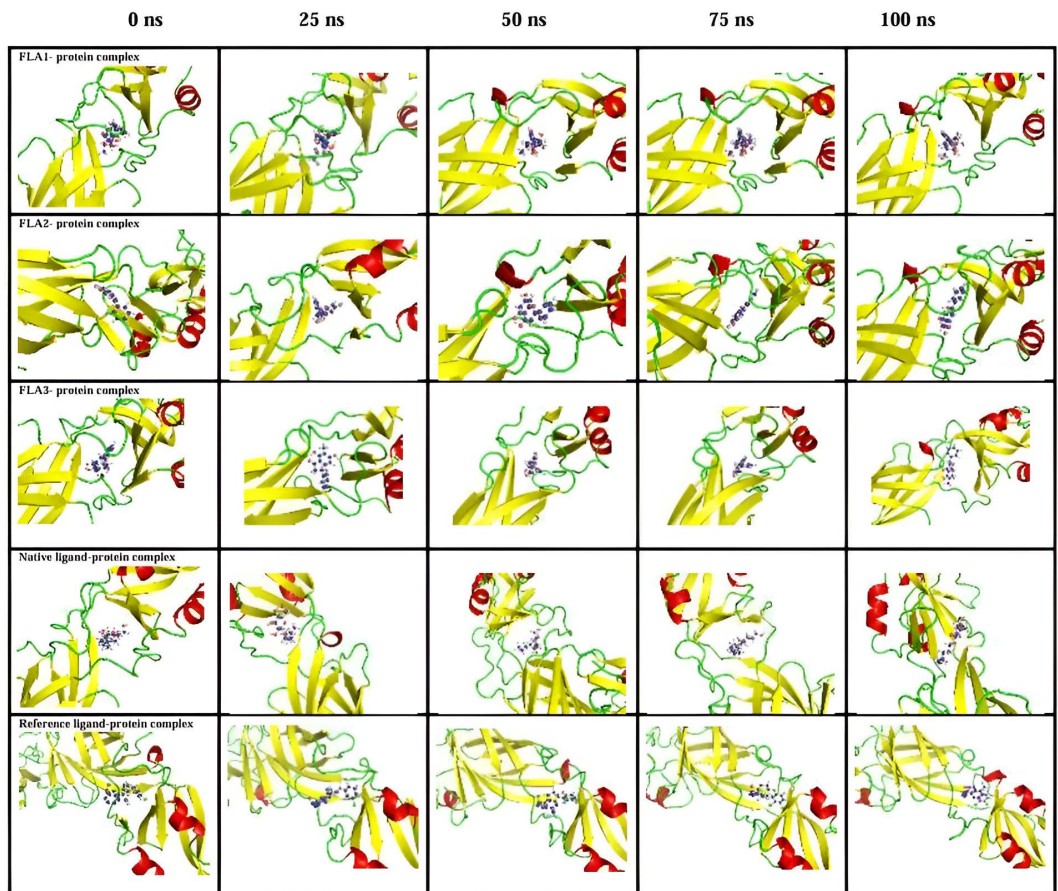

**Fig 6. Images capturing the ligand's positioning at the orthosteric site during MDS.** Orientation and location of ligands (shown in ball and stick model) to be monitored at various times.

by the large spikes in the protein backbone RMSD curve. However, after 30 ns, the protein backbone stabilized. The evolution of different molecules linked to a structural description was clearly understood due to their dynamic behaviour being constantly observed.

**3.5.8 PCA and DCCM analysis.** PCA and DCCM of the different complexes were performed through 100 ns MDS trajectories (S4 and S5 Figs). In PCA analysis, different colours (blue, white, and red) indicated the different stages (initial, intermediate, and final time steps) of the ligand's conformation in the simulation period [58]. In protein-FLA2, PC1 clusters had the most significant level of variability (56.94%), followed by PC2 clusters (16.69%) and PC3 (9.78%) clusters (S4 Fig). The FLA1-protein and FLA2-protein complexes show greater shifts in PC1, PC2, and PC3 than native ligand-protein and reference ligand-protein complexes. In addition, eigenvalue distribution also confirmed these tendencies, observing the significance of PC1 in describing major structural dynamics.

The DCCM plots depict the correlation effects of amino acids in MDS trajectories (S5 Fig). The FLA1-protein exhibited maximum interaction with the best distribution of positive correlations (0–1) in major functional regions and stability with reduced negative correlations (−1–0). Similarly, the FLA2-protein complex also displayed positive correlations in the active site but with higher anti-correlations, indicating a loss of flexibility. In contrast, the FLA3-protein complex displayed a mix of strong positive and negative correlations, which showed allosteric effects but with some indication of structural instability. Similarly, the native ligand-protein and reference ligand-protein complex displayed weak correlations and scattered anti-correlations, which indicates destabilization.

## 3.6. Binding free energy analysis

The binding free energy ($\Delta G_{BFE}$) and energy component analyses revealed protein-ligand interaction stability [59]. Among the ligands, FLA1-protein had the most favourable binding free energy (–29.1±5.83 kcal/mol) (except reference ligand –40.21±6.13 kcal/mol) than other protein-ligand complexes and protein-native complex (–18.94±3.2 kcal/mol) (Table 3). In the energy vs. frames plot, FLA1-protein exhibited the most stable interaction throughout the simulation with minimal fluctuations (S6 Fig).

Van der Waals energy (VDWAALS), electrostatic energy (EEL), polar solvation energy (EPB), and nonpolar solvation energy (ENPOLAR) are the main contributors to the total binding free energy (S7 Fig). Decomposition of the energy terms revealed that reference ligand-protein complex had favourable van der Waals' (–53.32±2.84 kcal/mol) and electrostatic (–24.49±10.15 kcal/mol) interactions with some penalty for desolvation ($\Delta E_{PB}$=43.17±5.57 kcal/mol) (S7 Fig). Residue decomposition analysis revealed that in the FLA1-protein complex, several active site residues namely THR48, GLU49, ALA50, LEU135, PHE193, LEU198, LEU207, GLN200, ILE270, GLN271, and LEU277, play a crucial role in the binding energy contributions. In contrast, for the reference ligand-protein complex, which exhibited a higher overall binding energy, the ligand interactions were primarily associated with VLA130, LEU198, ILE270, and LEU277 (S8 Fig). These findings were further supported by a heat map of complexes, where residue contributions are visually represented. In the heat map, darker blue colours correspond to residues with higher energy contributions. In contrast, lighter, faded colours indicated minimal involvement, confirming these residues' key roles in ligand binding (S9 Fig).

**Table 3. The components of binding free energy change ($\Delta G_{BFE}$: mean with standard deviation) for complexes.**

| Complexes | $\Delta E_{VDW}$ | $\Delta E_{EL}$ | $\Delta E_{PB}$ | $\Delta E_{NPOLAR}$ | $\Delta G_{GAS}$ | $\Delta G_{SOLV}$ | $\Delta G_{BFE}$ |
|---|---|---|---|---|---|---|---|
| **FLA1-protein complex** | −48.19±3.07 | −21.29±8.89 | 44.96±4.77 | −4.59±0.09 | −69.47±8.64 | 40.37±4.77 | −29.1±5.83 |
| **FLA2-protein complex** | −34.4±1.8 | −14.89±16.74 | 22.09±2.31 | −3.29±0.08 | −36.9±2.58 | 18.8±2.3 | −18.1±2.78 |
| **FLA3-protein complex** | −35.61±2.64 | −2.4±2.98 | 25.02±2.53 | −3.35±0.07 | −38±2.74 | 21.67±2.52 | −16.33±2.37 |
| **Native ligand-protein complex** | −26.51±2.89 | −13.01±7.52 | 24.09±6.64 | −3.5±0.21 | −39.52±5.20 | 20.59±6.55 | −18.94±3.2 |
| **Reference ligand-protein complex** | -53.32±2.84 | -24.49±10.15 | 43.17±5.57 | -5.58±0.13 | -77.8±9.76 | 37.59±5.52 | -40.21±6.13 |

## 4. Discussion

The geometrical and thermodynamic parameters suggested that FLA1 exhibited a strong and stable binding to the envelope protein, characterized by RMSD, Rg, SASA, H-bonding, RDF, PCA, DCCM, and free energy analysis throughout the 100 ns simulation period. These findings hinted that FLA1 may effectively inhibit the function of the envelope protein. Previous studies in which compounds with comparable docking scores targeted the same active site of the protein support this work [51,60]. However, those studies lacked detailed geometrical and thermodynamic analysis.

The promising performance of FLA1 showed potential as a lead compound for developing an antiviral drug when there is an urge for novel therapeutics due to the increasing cases of viral infection yearly. However, it is significant to recognize the limitations of a computational-only approach. While the *in silico* approach provides valuable insight into molecular interactions, stability, and toxicity, which helps to identify the potential target compound, they do not account for bioavailability, metabolic stability, or off-target effects, which are crucial for real-world efficacy. Therefore, further studies are required to validate *in silico* work, including *in vitro* assays to validate antiviral activity and cytotoxicity. Later, *in vivo* studies will evaluate pharmacokinetics and therapeutic potential. These steps are essential to translate the computational promise of FLA1 into a viable antiviral agent.

## 5. Conclusions

This study provides valuable computational data on the prospects of flavonoids as inhibitors of the dengue virus envelope (E) protein. FLA1 was identified to possess the highest binding affinity and stability among the compounds screened, confirmed through molecular docking and molecular dynamics simulations. Key stability parameters, including RMSD, RMSF, Rg, SASA, hydrogen bond count, and RDF, indicated a stabilized ligand-protein complex. Additionally, PCA and DCCM analysis revealed significant conformational stability and dynamic correlations, respectively, that also supported the effectiveness of FLA1. Binding free energy calculations (MM/PBSA) reconfirmed its tight binding to the target protein, while ADMET profiling suggested favourable pharmacokinetic properties. These findings designate FLA1 as a promising hit compound for dengue antiviral drug development, and its therapeutic efficacy must be confirmed with additional experimentation on *in vitro* and *in vivo*.

## Supporting information

**S1 Fig. Ramachandran plot of DENV-2 E protein (PDB ID: 1OKE).**
(TIF)

**S2 Fig. Molecular structure of top 3 flavonoids, Native ligand, and Reference ligand.**
(TIF)

**S3 Fig. Ligand docking position in the cavity with a hydrophobic surface (a) FLA1, (b) FLA2, (c) FLA3, (d) Native ligand, and (e) Reference ligand complexes.**
(TIF)

**S4 Fig. PCA analysis calculation of three eigenvalues (PC1, PC2, and PC3) for the (a) FLA1, (b) FLA2, (c) FLA3, (d) Native ligand, and (e) Reference ligand complexes.**
(TIF)

**S5 Fig. DCCM plots for the (a) FLA1, (b) FLA2, (c) FLA3, (d) native ligand, and (e) Reference ligand complexes.**
(TIF)

**S6 Fig. Changes in the binding free energy of different protein adducts with (a) FLA1, (b) FLA2, (c) FLA3, (d) Native ligand, and (e) Reference ligand, red indicates the moving average.**
(TIF)

**S7 Fig. MM/PBSA free energy contributions of different energies to the binding energy.** (a) FLA1, (b) FLA2, (c) FLA3, (d) native ligand, (e) Reference ligand, from the last 20 ns stable trajectories of the protein-ligand complexes.
(TIF)

**S8 Fig. MM/PBSA free energy partial contribution of the active amino acids and ligands to the binding free energy of (a) FLA1, (b) FLA2, (c) FLA3, (d) Native ligand, and (e) Reference ligand complexes.**
(TIF)

**S9 Fig. MM/PBSA the free energy of (a) FLA1, (b) FLA2, (c) FLA3, (d) Native ligand, and (e) Reference ligand complexes over the last 20 ns of the stable trajectory.** A heat-map showing residue-wise contributions per frame of the simulation.
(TIF)

**S1 Table. Toxicity from protox-III.**
(DOCX)

**S2 Table. Drug-Likeness Properties of hit candidates and reference drug through the Swiss ADME Server.**
(DOCX)

**S3 Table. ADMET properties from ADMETlab 2.0.**
(DOCX)

**S4 Table. Toxicity from ADMETlab 2.0.**
(DOCX)

**S5 Table. Toxicity from the pkCSM server.**
(DOCX)

## Acknowledgments

The authors would like to acknowledge the IT Innovation Centre, Tribhuvan University, Kirtipur, Kathmandu, Nepal.

## Author contributions

**Conceptualization:** Achyut Adhikari.

**Data curation:** Amar Waiba, Tika Ram Lamichhane, Madhav Prasad Ghimire.

**Formal analysis:** Amar Waiba, Anuraj Phunyal, Ashish Phuyal, Achyut Adhikari.

**Investigation:** Amar Waiba.

**Methodology:** Amar Waiba.

**Resources:** Amar Waiba.

**Software:** Amar Waiba.

**Supervision:** Achyut Adhikari.

**Validation:** Amar Waiba.

**Visualization:** Amar Waiba.

**Writing – original draft:** Amar Waiba, Anuraj Phunyal.

**Writing – review & editing:** Amar Waiba, Anuraj Phunyal, Tika Ram Lamichhane, Madhav Prasad Ghimire, Hari Nyaupane, Achyut Adhikari.

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
