## [Decision Letter · Decision Letter 0]

Dear Dr. Adhikari,

Thank you for submitting your manuscript to PLOS ONE. After careful consideration, we feel that it has merit but does not fully meet PLOS ONE’s publication criteria as it currently stands. Therefore, we invite you to submit a revised version of the manuscript that addresses the points raised during the review process.

**ACADEMIC EDITOR:**The study is appropriately designed and the composure of the manuscript is in order. However, the methodology needs to be presented comprehensively for reproducibility. For instance, the protein preparation steps currently exclude the generation of active binding pocket which is crucial for docking analysis. Secondly, how was the molecular docking analysis validated? What are the pharmacological implications of the molecular interactions observed with reference to the literature? All concerns of the reviewers also require substantial attention of the authors.. 

We look forward to receiving your revised manuscript.

Kind regards,

Yusuf Oloruntoyin Ayipo, Ph.D

Academic Editor

PLOS ONE

Additional Editor Comments:

The study is appropriately designed and the composure of the manuscript is in order. However, the methodology needs to be presented comprehensively for reproducibility. For instance, the protein preparation steps currently exclude the generation of active binding pocket which is crucial for docking analysis. Secondly, how was the molecular docking analysis validated? What are the pharmacological implications of the molecular interactions observed with reference to the literature? All concerns of the reviewers also require substantial attention of the authors.

Reviewers' comments:

Reviewer's Responses to Questions

Comments to the Author

1. Is the manuscript technically sound, and do the data support the conclusions?

Reviewer #1: No

Reviewer #2: Partly

2. Has the statistical analysis been performed appropriately and rigorously?

Reviewer #1: Yes

Reviewer #2: No

3. Have the authors made all data underlying the findings in their manuscript fully available?

Reviewer #1: Yes

Reviewer #2: Yes

4. Is the manuscript presented in an intelligible fashion and written in standard English?

Reviewer #1: Yes

Reviewer #2: No

Reviewer #1: This is a well-motivated and timely study that uses in silico approaches to investigate the potential of flavonoids as inhibitors of the dengue virus envelope (E) protein. The authors have clearly put substantial effort into employing a wide range of computational tools, including molecular docking, molecular dynamics simulations, ADMET profiling, principal component analysis (PCA), and MM/PBSA binding free energy calculations. While the overall methodology is appropriate and the results appear promising, the manuscript requires significant improvements in clarity, language, organization, scientific rigor, and interpretation in order to be suitable for publication.

• Ligand Selection: The criteria for selecting the 33 flavonoids are not clearly explained. Were these compounds chosen based on prior evidence of antiviral activity, structural diversity, or another rationale? Additionally, did the authors evaluate other flavonoids that failed to show positive results? Including this information would strengthen the justification for the selection.

• Docking Validation: The use of an RMSD threshold below 2 Å for docking validation is acceptable. However, a superimposed image of the native and re-docked ligand should be presented in the main manuscript rather than confined to the supplementary information.

• Simulation Parameters: Essential methodological details such as the timestep used, cutoff distances for van der Waals and Coulomb interactions, and the type of thermostat/barostat applied during equilibration must be included to ensure reproducibility.

• Table 2: The list of interacting amino acid residues should be refined by applying a 5 Å cutoff distance to exclude irrelevant or non-specific interactions.

• Interpretation of RMSD Values: While the authors emphasize RMSD plots, the manuscript lacks statistical comparisons or a discussion linking these fluctuations to functional or structural implications of ligand binding.

• Binding Energy Decomposition: The MM/PBSA energy breakdown is a valuable addition, but the manuscript would benefit from a more detailed biological interpretation. For instance, how do van der Waals and electrostatic interactions contribute to the binding affinity? Does FLA1 replicate or improve upon the interaction pattern of the native ligand βOG?

• Drug-likeness Criteria: The use of the term "drug-likeness" needs further support. It is not enough to rely solely on ADMET predictions: reference to standard frameworks such as Lipinski’s Rule of Five or Veber’s criteria should be included to substantiate the claim.

• Figures: The figure organization and presentation need significant improvement. All referenced figures must be embedded within the manuscript itself. For example, Figure 2 ("Two-dimensional projection of the protein-ligand interactions") is cited in the text but the actual image is missing. This undermines the clarity and completeness of the manuscript.

• Formatting and Writing Style: The authors should carefully review and align their manuscript with the PLOS ONE writing and formatting guidelines, including section organization, figure placement, and citation style. This will ensure the manuscript meets the journal’s structural and stylistic expectations.

• Broken Link: The manuscript contains an incorrect URL in the Methods section:

https://pubchem.ncbi.nlm.nhi.gov/

This is a typographical error and should be corrected to:

https://pubchem.ncbi.nlm.nih.gov/.

• Unmet Research Aim: In the Introduction, the authors claim:

“Therefore, effective treatment for dengue is still unknown which is the research gap of this study.”

However, the study does not actually propose or validate an effective treatment for dengue. Rather, it identifies a compound (FLA1) with promising in silico interaction potential. The manuscript does not address or close the research gap as stated, and this mismatch between the problem statement and the study’s outcome should be explicitly acknowledged and corrected.

Reviewer #2: This manuscript explored the potential inhibitory abilities of flavonoids against the dengue virus envelope (E) protein through computational analysis. Despite using a wide range of in silico techniques, such as molecular docking, MD simulations, ADMET screening, and MM/PBSA computations, the manuscript lacks the necessary research justification, methodological depth, and in-depth result analysis and interpretation required for publication at this stage.

1. Insufficient Justification of Study:

Although the introduction provides a broad overview of the dengue virus and the envelope (E) protein’s role in viral entry, the rationale for selecting flavonoids is poorly developed. The authors mention in general terms that flavonoids possess antioxidant and antiviral properties, but they do not explain why these specific 33 flavonoids were chosen or whether any of them have been previously associated with dengue or other viral infections. This is particularly important given that not all flavonoids have antiviral activity. Moreover, no supporting experimental evidence or literature references are provided to justify the inclusion of these compounds in the screening. Additionally, the overall purpose of the study is not clearly articulated, making it difficult to evaluate whether the study’s design aligns with its intended objectives.

2. Absence of a Proper/Standalone Discussion Section:

The current Results and Discussion section lacks depth and primarily presents data without insightful analysis or reference to existing literature. The biological significance of the results is not well captured. To strengthen the manuscript, I recommend that the authors include a standalone discussion section that addresses:

A. Interpretation of significant findings in the context of previous studies;

B. Potential implications for antiviral drug development;

C. Limitations of a computational-only approach;

D. Future directions such as in vitro/in vivo validation.

3. Insufficient Benchmarking:

Although the authors refer to n-octyl-β-glucoside (βOG) as the natural ligand targeting the DENV E protein binding pocket, its role is primarily as a site indicator rather than a functionally active antiviral compound. While βOG helps define the binding site in structural studies, it does not substitute for a control compound with validated antiviral potency. To strengthen the comparative framework, the authors should include or substitute known dengue virus E protein inhibitors or experimentally validated antiviral compounds as benchmarks.

4. Methodological Gaps and Lack of Adequate Statistical Analysis:

The authors assert ligand stability and therapeutic promise based on a single 100 ns molecular dynamics simulation. This is a short timescale, and no replicate simulations were performed to confirm reproducibility. Only conventional MD was conducted, without exploring enhanced sampling methods such as GaMD or metadynamics, which could provide more comprehensive conformational sampling. Additionally, there is no explanation for the choice of CHARMM27 over more updated force fields such as AMBER or CHARMM36m. The MM/PBSA computations, while present, are only superficially interpreted; there is no per-residue decomposition or analysis of specific energetic contributions from residues within the binding site.

5. Neurotoxicity and CNS Relevance Ignored:

Although CNS permeability (logPS < -3) is reported, the authors do not explain whether this is advantageous or detrimental. Since dengue virus is known to cause neurological complications, including encephalitis and meningitis, neurotoxicity is a relevant concern. It remains unclear whether BBB impermeability restricts the therapeutic potential of the screened compounds or if it is beneficial—for example, to prevent off-target CNS effects. The authors should explicitly address the CNS relevance of their findings, particularly in the context of drug safety and treatment efficacy.

6. ADMET Interpretation Is Superficial:

Although ADMET profiling is included, the manuscript offers minimal interpretation of the data. For example, toxicity class designations and plasma protein binding (PPB) values are reported without explaining their implications. What does a “good PPB” mean in the context of therapeutic safety or drug efficacy? Are the observed toxicity classes acceptable for antiviral use? These factors are essential for assessing the safety and drug-likeness of the candidates. Furthermore, there is no comparison between the flavonoid candidates and established antiviral medications. Conclusions like "CNS-impermeable" or "non-sensitizing" are given without elaborating on the significance of these characteristics for dengue treatment. ADMET results must be interpreted in a pharmacological context, not merely listed.

Overall Recommendation: Major Revision

To strengthen the manuscript, I recommend the authors address the following key areas:

Clearly articulate the aim and hypothesis, including justification for compound selection.

Add a comprehensive Discussion section.

Justify and refine the computational methodology, including force field choice and simulation protocols.

Replace or clarify benchmarking compounds, ensuring they have relevance to antiviral activity.

Discuss neurotoxicity and CNS relevance in the context of dengue pathology.

Provide deeper interpretation of ADMET and MM/PBSA results with appropriate comparative analysis.

**Do you want your identity to be public for this peer review?** For information about this choice, including consent withdrawal, please see our Privacy Policy

Reviewer #1: No

Reviewer #2: No

---

## [Author Response · Author response to Decision Letter 1]

1 Jun 2025

All the reviewers comments have been responded. Thank you.

---

## [Decision Letter · Decision Letter 1]

Computational insights into flavonoids inhibition of dengue virus envelope protein: ADMET profiling, molecular docking, dynamics, PCA, and end-state free energy calculations

PONE-D-25-15845R1

Dear Dr. Adhikari,

We’re pleased to inform you that your manuscript has been judged scientifically suitable for publication and will be formally accepted for publication once it meets all outstanding technical requirements.

Kind regards,

Yusuf Oloruntoyin Ayipo, Ph.D

Academic Editor

PLOS ONE

Additional Editor Comments (optional):

The study is timely and well-designed. Again, the submission meets the level of scientific rigour required for publication in this title and all the concerns raised by the respective reviewers have been addressed satisfactorily. I hereby recommend the manuscript for publication in the current version.

Reviewers' comments:

Reviewer's Responses to Questions

Comments to the Author

Reviewer #1: All comments have been addressed

Reviewer #2: All comments have been addressed

2. Is the manuscript technically sound, and do the data support the conclusions?

Reviewer #1: Yes

Reviewer #2: Yes

3. Has the statistical analysis been performed appropriately and rigorously?

Reviewer #1: Yes

Reviewer #2: Yes

4. Have the authors made all data underlying the findings in their manuscript fully available?

Reviewer #1: Yes

Reviewer #2: Yes

5. Is the manuscript presented in an intelligible fashion and written in standard English?

Reviewer #1: Yes

Reviewer #2: Yes

Reviewer #1: The authors have satisfactorily completed the requested revisions. The manuscript is now ready for acceptance

Reviewer #2: The revised manuscript has adequately addressed the major concerns raised in the initial review. I do not have additional concerns and support the acceptance of this manuscript in its current form.

**Do you want your identity to be public for this peer review?** For information about this choice, including consent withdrawal, please see our Privacy Policy

Reviewer #1: No

Reviewer #2: No

---

## [Editor Report · Acceptance letter]

PONE-D-25-15845R1

PLOS ONE

Dear Dr. Adhikari,

I'm pleased to inform you that your manuscript has been deemed suitable for publication in PLOS ONE. Congratulations! Your manuscript is now being handed over to our production team.

Kind regards,

on behalf of

Dr. Yusuf Oloruntoyin Ayipo

Academic Editor

PLOS ONE